# Microbiome in Nasal Mucosa of Children and Adolescents with Allergic Rhinitis: A Systematic Review

**DOI:** 10.3390/children10020226

**Published:** 2023-01-27

**Authors:** André Costa Azevedo, Sandra Hilário, Micael F. M. Gonçalves

**Affiliations:** 1Department of Pediatrics, Unidade Local de Saúde do Alto Minho, 4904-858 Viana do Castelo, Portugal; 2CESAM, Department of Biology, University of Aveiro, 3810-193 Aveiro, Portugal; 3Division of Microbiology, Department of Pathology, Faculty of Medicine, University of Porto, 4200-319 Porto, Portugal

**Keywords:** allergic rhinitis, chronic disorder, dysbiosis, microbiome, pediatric

## Abstract

The human upper respiratory tract comprises the nasal cavity, pharynx and larynx regions and offers distinct microbial communities. However, an imbalance and alterations in the nasal mucosa microbiome enhance the risk of chronic respiratory conditions in patients with allergic respiratory diseases. This is particularly important in children and adolescents once allergic rhinitis (AR) is an inflammatory disorder of the nasal mucosa, often associated with an increase in pulmonary allergic inflammation. Therefore, this systematic review aimed to collect scientific data published concerning the microbial community alterations in nasal mucosa of children and adolescents suffering from AR or in association with adenotonsillar hypertrophy (AH) and allergic rhinoconjunctivitis (ARC). The current study was performed using the guidelines of Preferred Reporting Items for Systematic Reviews and Meta-Analyses (PRISMA). Publications related to microbiome alterations in the nasal mucosa in pediatric age, studies including next-generation sequencing platforms, and studies exclusively written in the English language were some of the inclusion criteria. In total, five articles were included. Despite the scarcity of the published data in this research field and the lack of prospective studies, the genera *Acinetobacter*, *Corynebacterium*, *Dolosigranulum*, *Haemophilus*, *Moraxella*, *Staphylococcus* and *Streptococcus* dominate the nares and nasopharyngeal microbiome of the pediatric population regardless of their age. However, an imbalance in the resident bacterial community in the nasal mucosa was observed. The genera *Acinetobacter*, and *Pseudomonas* were more abundant in the nasal cavity of AR and AH children, while *Streptococcus* and *Moraxella* were predominant in the hypopharyngeal region of AR infants. An abundance of *Staphylococcus* spp. was also reported in the anterior nares and hypopharyngeal region of children and adolescents suffering from AR passive smoke exposure and ARC. These records suggest that different nasal structures, ageing, smoke exposure and the presence of other chronic disorders shape the nasal mucosa microbiome. Therefore, the establishment of adequate criteria for sampling would be established for a deeper understanding and a trustworthy comparison of the microbiome alterations in pediatric age.

## 1. Introduction

Recent research has shown that the human microbiome has a pivotal role in health and disease pathogenesis [1]. This has previously been demonstrated in the gastrointestinal tract [2] and lower airways (lung) [3] but has not been thoroughly investigated in the upper airways (nasal and oral passages).

The human nasal mucosa harbors a high number of microorganisms, including commensal microbes that maintain the stability of their microenvironment in a symbiotic relationship [4,5]. However, reactive nasal inflammatory conditions such as allergic rhinitis (AR) can cause a disruption in nasal microbiota (dysbiosis) and deeply impact human health, by enhancing the prevalence of chronic respiratory disorders (e.g., asthma, and chronic obstructive pulmonary disease) [6]. AR is a frequent respiratory disorder of the nasal mucosa in the pediatric population characterized by sneezing, rhinorrhea and nasal congestion, whose prevalence is increasing every year worldwide [7,8].

Accumulating evidence suggests that the microbiome in nasal mucosa may hold an important function in the alteration of immune responses and the development of AR [4]. With the advance of high-throughput sequencing techniques, metagenomics has represented an important approach that can not only predict the abundance and richness of the microbial community in the nasal mucosa, but also reveal the role of the microbiome in health and disease [5,9,10]. By applying such sequencing methods, some researchers have shown data suggesting dysbiosis of the nasal microbiota in adults regarding allergic inflammation of the airways [7,9,11,12]. For instance, Lee et al. [13] reported that *Corynebacterium*, *Propionibacterium*, *Staphylococcus*, and *Streptococcus* are common bacterial genera in the nasal mucosal diseases of AR, compared to healthy controls. Moreover, Yau et al. [14] demonstrated that the genus *Staphylococcus* is more abundant in nasal mucosa of patients with allergic rhinoconjunctivitis (ARC). 

In children and adolescents, most of the research has been conducted mainly towards the role of microbiota in the lower airways such as asthma [1,15,16]. On the contrary, the alterations in nasal mucosa associated with AR in childhood and adolescence has been somehow neglected and not studied in detail [8,12,14,17,18]. The few studies concerning the nasal microbiome composition in children with AR have reported, for instance, an abundance of members of the genus *Moraxella* [19]. Chiu et al. [19] pointed out that *Moraxella* spp. may be involved in controlling allergen sensitization in AR. Moreover, it has been demonstrated that early-life microbiota of nasal mucosa is linked with subsequent development of respiratory tract infections and rhinitis [6,20,21]. Therefore, the microbiome of the upper airways becomes a potential source for researchers to understand the linkage between chronic respiratory diseases and the microbiome composition [1,6,21]. 

However, there is still limited research into the relationship between nasal microbiome dysbiosis and the development of AR in children and adolescents. For this reason, it is crucial to elucidate the broad spectrum of the nasal mucosa microbiome alterations for identifying potential biomarkers for AR diagnosis and effective treatment [21]. This is particularly important in children, given that AR has been often mistreated and can aggravate existing asthma [1,22]. Therefore, this study aims to be an up-to-date systematic review focused on summarizing the current data for associations between AR and the nasal mucosa microbiome in children and adolescents. 

## 2. Materials and Methods

The current systematic review was performed using the PRISMA guidelines (Preferred Reporting Items for Systematic Reviews and Meta-Analyses) [23]. The methodology used, including the screening of the title and abstract, full text reading, inclusion/exclusion and data extraction criteria were accepted by all authors. A review protocol was registered to PROSPERO, a database of systematic review protocols (registration number: ID 390751).

### 2.1. Search Strategy

We performed an electronic search on 4 databases, MEDLINE (PubMed), Web of Science, Scopus, and Google Scholar on 20 October 2022. Publications were also manually checked for additional cited references. The scientific review was performed with no limitations to language, year of publication and publication type. To ensure that the search is inclusive, keywords (alone or in combination) related to scientific literature regarding the microbiome of nasal mucosa of children and adolescents with AR were included. For this research, we used the MeSH Browser from the NIH (National Institute of Health), which allowed us to search directly for MeSH terms. The following query was used: (Microbiota OR Microbial Community OR Community, Microbial OR Microbial Communities OR Microbial Community Composition OR Community Composition, Microbial OR Composition, Microbial Community OR Microbial Community Compositions OR Microbial Community Structure OR Community Structure, Microbial OR Microbial Community Structures OR Microbiome OR Microbiomes OR Human Microbiome OR Human Microbiomes OR Microbiome, Human) AND (Child OR Children OR Preschool Child OR Children, Preschool OR Preschool Children OR Childhood OR Adolescent OR Adolescents OR Adolescence OR Teens OR Teen OR Teenagers OR Teenager OR Youth OR Youths OR Adolescents, Female OR Adolescent, Female OR Female Adolescent OR Female Adolescents OR Adolescents, Male OR Adolescent, Male OR Male Adolescent OR Male Adolescents OR infant OR infants OR Infant, Newborn OR Newborns OR Newborn OR Neonate OR Neonates OR Pediatric OR Pediatrics) AND (Rhinitis OR Rhinitides OR Allergic Rhinitides OR Allergic Rhinitis) AND (Nose OR Nasal OR Nasal Mucosa OR Nasal Epithelium OR upper-airway OR upper airway OR upper-airways OR upper airways OR Sinonasal OR Sinus OR Sinuses OR Sinonasal Tract OR Sinonasal Tracts OR Rhinopharynxes OR Rhinopharynges OR Rhinopharynx OR Nasopharynges OR Nasopharynxes OR Choanae OR Turbinate). The studies exported from the databases were uploaded to Covidence systematic review manager for further selection and quality assessment.

### 2.2. Inclusion and Exclusion Criteria

This systematic review was performed to be an updated review and to summarize and gather the current evidence for microbial dysbiosis in the nasal mucosa of children and adolescents with AR. The inclusion criteria were selected to make the review much more adaptable and relevant to the research question. Articles meeting the following criteria were eligible for inclusion: (1) studies concerning any change in the nasal microbiome in pediatric age associated with AR; (2) assessment of diversity and composition of the nasal microbiome using culture-independent techniques for whole spectrum of detected microbiota. These techniques include next-generation sequencing platforms such as pyrosequencing, HiSeq, MiSeq, and whole metagenome sequencing; (3) studies with AR in pediatric age with control groups, coupled by adequate statistical analyses; (4) studies using nasal samples, including anterior nares, nasal cavity, sinuses, rhinopharynx, nasopharynx swabs or fluid aspirates. The included articles were also selected with a special focus on publications from the last 5 years (2017–2022), which represents more than 78% of the references used (33 out of 42).

Articles highlighting microbiome composition studies in participants with respiratory diseases other than AR and those targeting small sample size such as less than 10 participants were excluded. This systematic review aimed to include original articles without restrictions regarding the type of classification of the studies. Thus, studies whose outcomes could not be of interest for this work, due to being self-reported or not using objective measures, were excluded. This included reviews, poster presentations, book chapters, theses and dissertations, conference papers, letters, editorials, case reports, articles with only abstract available, studies with culture-dependent techniques, and animal studies. Non-English articles were also not considered, to avoid introducing bias in this review or ignoring key data due to misunderstanding translations.

### 2.3. Study Selection

The above-mentioned search strategy was used to identify studies eligible for full-text screening. Titles and abstracts of the studies retrieved were firstly screened independently by two authors (A.C.A and S.H.). The eligible full texts were selected and independently assessed by the authors mentioned above. Any divergences concerning the eligibility of studies were discussed with a third author (M.F.M.G.). The identified studies were subsequently used for data extraction and quality assessment.

### 2.4. Data Extraction

Data from the included studies were extracted for assessment of study quality. The following information was collected from each included study: author, study design, study objectives, number of participants, age, study setting (including ethnicity and country), methodology (including type of microbiome analysis and sample type method), key findings and limitations.

### 2.5. Study Quality Assessment

The quality of the included studies was assessed using the NIH Study Quality Assessment Tools for observational studies [24] and Cochrane risk-of-bias tool for randomized trials [25]. The assessment was achieved by A.C.A and S.H. Divergences resulting from this process were discussed with the third author (M.F.M.G.).

## 3. Results

### 3.1. Number of Retrieved Papers

The literature search generated 144 articles, resulting in 102 unique articles after removal of duplicates. After title and abstract screening, 90 articles were excluded based on title and study type. The full text of the remaining 12 articles eligible were assessed, from which 7, not meeting the inclusion criteria mentioned above, were excluded (adult population, *n* = 4; non-study target area, *n* = 2; non-English articles, *n* = 1). A flowchart of the search strategy and study selection process of the articles is shown in Figure 1.

The characteristics and taxonomic findings of the five studies finally selected are summarized in Table 1. Two studies assessed and evaluated the bacterial community in the nasal mucosa of AR infants. One study only collected fluid samples from the hypopharyngeal region through the nose in infants at 1 week, 1 month, and 3 months of life and evaluated the outcome of AR at 6 years of age. Another study obtained samples from the anterior nostrils in infants, then followed up and assessed the outcome of AR and wheeze at 3 weeks and 3, 6, 9, 12, 15, and 18 months of age. The remaining three studies were performed in pediatric patients aged 6–18 years, using samples from the nasal nostrils and the nasopharyngeal regions, but with different outcomes. These studies aimed to evaluate the influence of passive smoke on the nasal microbiome of children with AR, to assess the microbial composition in subjects with AR, AH or both diseases, and to evaluate the alterations in nasal and ocular microbiome in patients with ARC. All studies included in this review employed 16S rRNA gene amplicon sequencing to assess taxonomic composition. However, the amplified regions of the 16S rRNA gene were different (V3–V4, *n* = 2; V4, *n* = 1; V3–V6, *n* = 1; and without information, *n* = 1). Fungal and viruses’ fraction of the microbiota were not analyzed in these studies.

### 3.2. Microbiome Diversity and Richness

The changes in bacterial community structure and composition in AR children were measured in all studies hereby included. Species richness and bacteria taxonomic diversity was calculated as a measure of alpha-diversity by the number of observed Operational Taxonomic Units (OTUs) and Shannon/Simpson’s indexes, respectively. Alpha-diversity was lower in infants with AR than in healthy control subjects [17,18]. Decreased bacterial richness (low diversity) was first reduced at age 1 month, and then at 3 months in infants developing asthma by 6 years [17]. Additionally, Ta et al. [18] observed that microbiome diversity in infants with AR with and without wheeze decreased over time in comparison with healthy children (HC) in the first 18 months. Marazzato et al. [12] found no statistically significant differences between HC children at age 6–12 years and those with AR. Despite this, it seems evident that the passive smoke in children with AR highly influenced the nasal microbiome. Brindisi et al. [8] showed lower values of biodiversity in children with AR exposed to secondhand smoke than in children with AR not exposed. In contrast, in another study, the microbiome diversity in the nasopharyngeal region of ARC subjects aged 6–18 years was observed to be higher than HC [14].

Different metric measures (e.g., Bray–Curtis, and weighted-UniFrac) were used for beta-diversity calculation (variation in the composition of the microbial community within the samples). Distinct bacterial community composition was observed between HC and AR subjects at all ages. However, in the study of Ta et al. [18], which characterized the nasal microbiota of children with AR and wheeze during the first 18 months, the authors observed that these differences in microbiota composition between the groups were less distinct after 6 months of age.

### 3.3. Taxonomic Composition

The main changes in nasal mucosa microbiome composition and associations between HC and AR subjects in pediatric age are summarized in Figure 1. As Brindisi and co-authors [8] do not provide the raw abundance data, the figure does not include this study.

#### 3.3.1. Nasal Cavity Microbiota

Three studies investigated the alterations in the taxonomic compositions in the nasal cavity.

Regarding phylum level, Marazzato et al. [12] revealed that in children aged 6–12 years old, Actinobacteria, Firmicutes and Proteobacteria were the most abundant in HC and AR children. In AR subjects, Actinobacteria showed a decrease comparing to HC, accompanied by an increase in Proteobacteria. These results were also observed in Ta et al. [18], who demonstrated a larger reduction in Actinobacteria in subjects with AR with wheezing compared to those without wheezing. 

At the genus level, Marazzato et al. [12] showed that Corynebacterium and Moraxella followed by Pseudomonas and Acinetobacter were the most abundant genera in HC. On the other hand, Pseudomonas and Acinetobacter demonstrated the higher relative abundance in AR subjects, with a considerable reduction in Moraxella and Corynebacterium. Ta et al. [18], indicated that Corynebacterium and Alloiococcus were the most abundant genera in toddlers in the first 18 months of life, and no significative differences were found in their relative abundance comparing HC to AR subjects with or without wheeze. Other genera found were Staphylococcus, Moraxella, Streptococcus and Haemophilus. 

Brindisi et al. [8] compared the nasal microbiome of children aged 6-16 years with AR exposed to secondhand smoke with those not exposed to smoke. The results indicated that in non-exposed AR children, five species belonging to the genus Pseudomonas were found, followed by three species of each of the following genera: Moraxella, Serratia, Staphylococcus, Streptococcus and Corynebacterium. Moraxella nonliquefaciens was the most abundant species found in non-exposed AR children in contrast to exposed AR children, which is in line with Marazzato et al. [12], who also found that this species was the most abundant in the nasal cavity in children 6 to 12 years old. In AR children exposed to smoke, Staphylococcus epidermidis and Serratia quinivorans were the most abundant species. Contrarily, such a result was not verified in non-exposed AR children, where these species were less expressive.

#### 3.3.2. Nasopharyngeal Microbiota

Yau et al. [14] evaluated the alterations in nasal and ocular microbiota of children with ARC. At the phylum level, Actinobacteria, Firmicutes and Proteobacteria were the most abundant in nasopharyngeal region in HC. Comparing the relative abundance of these phyla with ARC subjects, it a decrease in the abundance of these phyla was evident, which was also accompanied by an increase in other phyla including Bacteroidetes, Fusobacteria, Deinococcota and Cyanobacteria.

At the genus level, Moraxella, Corynebacterium, Dolosigranulum and Haemophylus were the most common genera found in HC subjects. In ARC subjects, a decrease in the relative abundance of Moraxella, Corynebacterium and Haemophilus and an increase in Staphylococcus and Streptococcus was observed.

#### 3.3.3. Hypopharyngeal Microbiota

Morin et al. [17] evaluated the development of AR at age 6 years based on early-life microbial exposures. The most common genera found in HC at the first 3 months of life were *Streptococcus* and *Moraxella* followed by *Staphylococcus*. In children that later developed AR, *Streptococcus* and *Moraxella* remained the most abundant genera. However, the relative abundance of *Staphylococcus* decreased, and *Corynebacterium* increased. 

## 4. Discussion

This is the first systematic review that evaluates the microbiome of nasal mucosa of children and adolescents with AR.

Alpha diversity in microbiome studies is defined as the mean diversity of species in different sites at a local scale. It can be assessed using, for instance, Pielou, Shannon, and Simpson indices [26]. A study conducted by Chiang et al. [11] reported that Pielou and Simpson indices showed a significant decrease in the nasal extracellular vesicles of AR patients aged 15–36 years. Although the studies carried out by Morin et al. [18] and Ta et al. [18] used younger subjects and have quantified the richness diversity by the Shannon diversity index, the results corroborated those from Chiang et al. [11]. While Morin et al. [17] reported a decrease in bacterial diversity in the hypopharyngeal region of children aged 1 and 3 months, Ta et al. [18] observed that microbiome diversity in the nasal cavity of infants in the first 18 months with AR decreased, regardless of the presence or not of wheeze symptoms, when compared to HC. Such an outcome is indicative that microbial diversity decreases in AR children, and such microbial dysbiosis might be linked with the increased sensitization to allergic disease [7,11,19,27]. 

This study showed that there is very limited knowledge on the nasal mucosa microbiome in pediatric age, mainly towards microbiota alterations associated with AR [8,12,14,17,18]. Despite the limited studies, several bacterial taxa have been indicated as resident microbiota of the nasal mucosa in children and adolescents, such as *Acinetobacter*, *Corynebacterium*, *Dolosigranulum*, *Haemophilus*, *Moraxella*, *Staphylococcus* and *Streptococcus*. This suggests that these taxa may behave as commensal bacteria in nasal mucosa, which is of the utmost importance once a healthy and balanced microbiota confer protection over pathogens [12,28]. Nevertheless, resident commensal bacteria may behave as pathogenic organisms when they are exposed to environmental changes (e.g., allergen sensitization). This causes a disruption in the nasal microbiota which favors the development of an advantageous microenvironment for the pathogens to stablish [4,6,7,12]. 

The nasal mucosa microbiome in childhood and adolescence in cases of AR provided in Figure 2 indicates that different bacteria can colonize various structures of the upper respiratory tract. These include the nasal cavity and two pharyngeal regions, the nasopharynx and hypopharynx (laryngopharynx). After analyzing the nasal microbiome represented in this review, it was evident that some bacterial communities are site specific. For instance, the genus *Alloiococcus* was one of the most abundant and detected only in the anterior nares of AR and HC infants in the first 18 months [18]. The authors have stated that *Alloiococcus* seems to be related with rhinitis and wheeze. On the contrary, Teo et al. [29] reported an abundance of the genus *Alloiococcus* in nasopharyngeal samples from healthy infants aged 2–12 months. This suggests that this genus may be a common component of a balanced healthy nasopharyngeal region as well as it may enhance the microbiome stability of the nasopharynx [29]. Nevertheless, *Alloiococcus otitidis* is the only known species from the genus, and it is often detected in the middle ear and nasopharyngeal region of children prone to develop acute otitis [30]. For this reason, there is a lack of evidence to clarify the role of the genus *Alloiococcus* in the nasal cavity in infants, as well as whether *A. otitidis* is a commensal bacterium or a pathogen [31].

The microbiome of the nasopharyngeal region in 2–12-month-old children is dominated mainly by the genera *Moraxella*, *Streptococcus*, *Corynebacterium*, *Staphylococcus*, *Streptococcus*, *Haemophilus* and *Dolosigranulum* [29]. Despite the study by Yau et al. [14] using older subjects, all these genera were detected in the nasopharynx microbiome, although with different abundances considering whether the samples are from HC or AR children. The genera *Moraxella*, *Dolosigranulum* and *Corynebacterium* were the most abundant genera present in the nasopharyngeal region in HC aged 6–12 years [14]. Such an outcome is corroborated by Folino et al. [32], who collected nasopharyngeal swabs from HC aged 2–4 years and found *Dolosigranulum* and *Corynebacterium* as the most abundant genera. This suggests that these bacterial genera may represent a pivotal role in the nasal mucosa microbiome of HC [12]. However, further studies should be conducted to understand this role in regulating the respiratory tract of children [32].

Marazzato et al. [12], Yau et al. [14] and Morin et al. [17] demonstrated in their studies that *Moraxella* was one of the main genera commonly present in the nasal cavity, nasopharyngeal and hypopharyngeal regions of healthy participants, regardless of their age. In fact, recent research has provided evidence about the potential role of the species *M. nonliquefaciens* associated with a healthy nasal microbiota in children [12,33]. Although this species was also found in the nasopharynx of 4–7-year-old HC, it has been associated with a higher risk of developing acute sinusitis and lower airway infection [34]. In line with this, Chiu et al. [19] collected throat swabs samples from children aged 3–5 and reported that the genus *Moraxella* was found to be more abundant in children with AR than in HC. The authors have also highlighted that *Moraxella* spp. may possibly cause susceptibility to asthma and rhinitis.

All studies included in this systematic review performed the 16S rRNA sequencing to predict the diversity and abundance of the bacterial community in the nasal mucosa but focusing on different objectives. Marazzato et al. [12], Yau et al. [14] intended to evaluate the microbial composition of the nasal mucosa in subjects suffering from a chronic disorder such as AR, AH, and ARC. However, while Brindisi et al. [8] evaluated the influence of an environmental factor (passive smoke exposure) on the nasal microbiome of children with AR, Ta et al. [18] and Morin et al. [17] assessed the outcome of AR in the first months of life, and how early-life microbial exposures can influence subsequent respiratory infections at older ages.

Early-life microbiota of nasal mucosa is associated with subsequent development of respiratory tract infections and rhinitis [20]. Moreover, it has been suggested that the presence of siblings, male sex or cesarean delivery may be important determinants for dysbiosis in the nasal microbiota [18]. Teo et al. [29] showed that the genus *Streptococcus* was the second most abundant in healthy infants aged 2–12 months. These authors demonstrated that the early colonization by *Streptococcus* at 2 months of age, resulted in a significant association with persistent wheeze at 5 years of age. In line with this is the most recent study of Morin et al. [17], who reported the presence of the genus *Streptococcus* in the hypopharyngeal region in 1–3-month-old infants regardless of suffering or not from AR. The authors have also concluded that microbial diversity of the airways at the early life is prone to increase the risk of children developing AR by age 6 years. Thus, it is suggested that the early *Streptococcus* colonization may increase the risk of developing asthma, as previously advocated by Teo et al. [29]. Bisgaard et al. [20] also demonstrated that the colonization of *Haemophilus influenzae*, *M. catarrhalis* and *S. pneumoniae* in the throats of 1-month-old infants was associated with chronic wheeze in the first 5 years. Additionally, Vissing et al. [35] assessed the nasal microbiome through hypopharyngeal aspirates in infants at 4 months and suggested that neonatal colonization of the airways with *Moraxella* spp. increased the risk of bronchiolitis and pneumonia in the first 3 years of life. Moreover, Ta et al. [18] observed differences in microbiota composition profiles of infants with AR aged 3 weeks and 3 months, in those who developed wheeze in the first 18 months of life. 

Pediatric passive smoke exposure (PSE) reaches up to 40% of children [36]. PSE may lead to a microbial dysbiosis, thus increasing the risk of chronic inflammation of nasal mucosa of AR children [8] or increasing respiratory diseases including asthma, bronchitis, and pneumonia [37]. However, to our knowledge, only a few reviews have reported the association between PSE and AR in children [36,38]. In this systematic review, we highlighted a study carried out by Brindisi et al. [8], who aimed to evaluate the influence of smoke exposure on the nasal mucosa microbiome composition in children with AR aged 6–16 years. The authors revealed that PSE influenced the nasal mucosa microbiome, by decreasing the bacterial community in children suffering from AR. Moreover, it was also shown by Brindisi et al. [8] that nasal obstruction worsens in AR children presenting higher cotinine levels (nicotine metabolite). This is corroborated by Shargorodsky et al. [36], who pointed out that tobacco smoke exposure might increase the prevalence of rhinitis symptoms. Moreover, the presence of *Staphylococcus epidermidis* was positively correlated with cotinine levels, suggesting that this species may adapt to smoke exposure conditions [8]. This is in line with Wu et al. [39], who proved that nicotine enhances the development of *S. epidermidis* biofilms, promoting its establishment in the mucous membranes such as the nasal mucosa.

Comparing the present results with a recent review by Tai et al. [5] of AR adults, the taxa present are similar to some of the most prevalent taxa found in children, such as the genera *Staphylococcus* and *Corynebacterium*. The authors also suggested that *S. aureus* are present in the nasal mucosa of patients with AR in a considerably higher abundance compared to nasal mucosa of healthy individuals. In agreement with this study, Kim et al. [40] also demonstrated that *S. aureus* exhibited the highest abundance in patients with AR. For this reason, this species may contribute to the pathogenic mechanism of AR, inducing the release of type 2 interleukins [5,40]. Furthermore, Gan et al. [7] revealed that the abundance of the genus *Pseudomonas* in patients aged 18–91 years was higher in those suffering from AR than in healthy controls, which is in line with Marazzato et al. [12], who also found an increase in this genus. In addition, a decrease in *Haemophilus* abundance in AR patients was observed in Gan et al. [7], which corroborates the findings of Yau et al. [14].

Therefore, these findings represent an important milestone in understanding how the microbiome influences the development of allergic diseases, particularly AR. Moreover, it may be the first step to develop detection strategies in order to intervene precociously.

We recognize that this review poses some limitations. First, the sample size of four studies [8,12,14,18] is small (ranging from 13 to 60 subjects), with the exception of one study out of five that comprised up to 407 HC participants [17]. Such limitations can hamper the confirmation of differences in microbiome diversity and composition between AR and HC patients. Additionally, it is recognized that nasal bacterial diversity may be influenced by living environment, in which air pollution can cause various respiratory diseases [41]. Two of the studies hereby included focused on participants from Asian countries: Singapore [18] and China [14]. Considering that the exposure to environmental pollutants is increasing in Asia [42], the results of microbial communities from these participants may bias the results, hampering an accurate comparison between studies. 

Patients with AR and asthma are often treated with topical and systemic treatments that can affect the resident microbiome [10]. Therefore, we recognize that nasal microbiota may be affected by the pharmacotherapy used in pediatric subjects suffering from AR such as antihistamines or intranasal corticosteroids [22].

Only one study faced difficulties in the identification of all bacterial groups at the species level [18], thus the results from this study must be carefully analyzed. Moreover, another limitation in this study was the different targeted sample types from which it is unknown whether they were collected or not by doctors with pediatric training: the anterior nares [8,12,18], the nasopharyngeal [14] and the hypopharyngeal [17] regions. The analysis of different nasal regions turns the comparison of results a prevailing challenge, given the differences in bacterial composition between different structures of the respiratory tract.

## 5. Conclusions

This systematic review is the first to report the alterations in the nasal mucosa microbiome in children and adolescents suffering from AR. The analysis of the selected studies provides evidence of limited knowledge on this research field in pediatric age. Nevertheless, the available articles mentioned in this review showed that the bacterial community in the nasal mucosa microbiome tend to decrease in subjects with AR, compared to HC. This suggests that a dysbiosis in nasal mucosa may impact children health, by rising the prevalence of chronic respiratory conditions (e.g., asthma, pneumonia). These studies are particularly important, to help clinicians obtain better knowledge of nasal microbiome and allergic inflammation, as well as offer insights for future development of potential biomarkers for AR diagnosis and its treatment.

In future studies, the establishment of adequate criteria for sampling methods would be favored and of the utmost importance for a deeper knowledge of the nasal microbiome in pediatric age. These criteria should include knowledge on the:(1)ethnical records of the individuals;(2)allergic predisposition records;(3)environmental factors;(4)pharmacotherapy used;(5)anatomical variants (e.g., adenotonsillar hypertrophy or nasal septal deviation).

Moreover, future studies targeting the role of the early-life microbiome in the development of respiratory infections later in life should also take into consideration several determining factors, mainly to assess the gender of subjects, cesarean delivery, existence of siblings, and infant care attendance, once these criteria influence the establishment and development of the nasal microbiome.

## Figures and Tables

**Figure 1 children-10-00226-f001:**
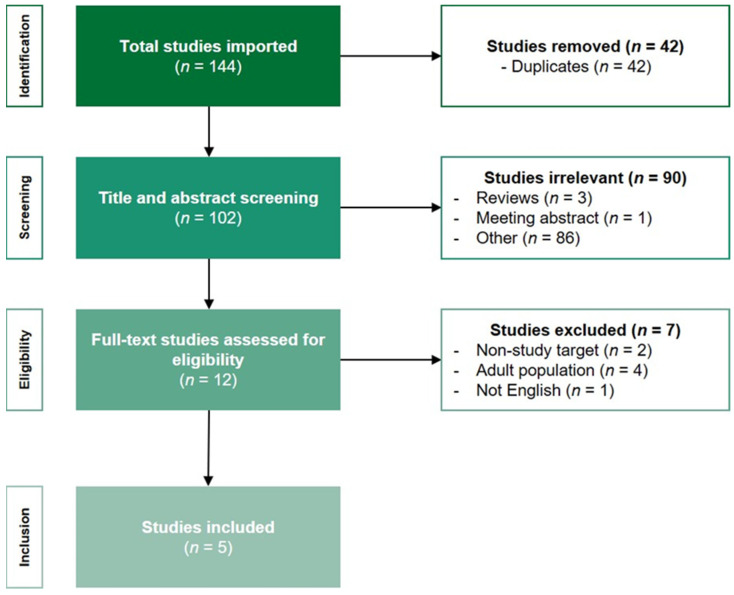
Flowchart of the search strategy.

**Figure 2 children-10-00226-f002:**
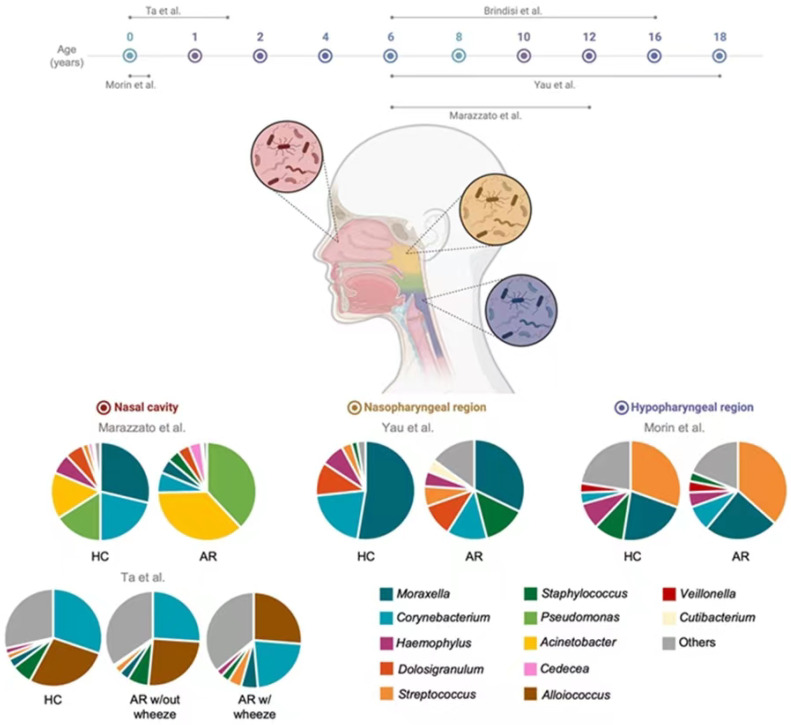
Graphic representation of the composition of the nasal mucosa microbiome in childhood and adolescence in cases of AR. This figure was based on studies that reported the abundance on genus level. AR: Allergic rhinitis; HC: healthy children. The figure was created with BioRender.com (accessed on 26 November 2022). References [8,12,14,17,18] are cited in the figure.

**Table 1 children-10-00226-t001:** Overview of the final selected articles and the characteristics and reported findings. AR: Allergic rhinitis; PSE: passive smoke exposure; AH: adenotonsillar hypertrophy; HC: healthy children; DNAm: DNA methylation; ARC: Allergic rhinoconjunctivitis; ↑ increase; ↓ decrease.

Author	Study Design	Study Objective	Participants	Age	Ethnicity Country	Sample Type	Microbial Analysis	Key Findings	Limitations	Quality Score
Brindisi et al. [8]	Cross-sectional	Influence of passive smoke on the nasal microbial composition in children with AR	AR: 50 (25 PSE vs. 25 negatives)	6–16 years	Unknown, Italy	Anterior nasal swab	16S rRNA amplification	Children with a PSE: ↓ biodiversity and shift in the microbiota composition. AR PSE nasal microbiota: abundance of *Staphylococcus epidermidis* and *Serratia quinivorans*. AR Non-PSE nasal microbiota: abundance of *Moraxella nonliquefaciens*.	Small sample size. Monocentric evaluation. No ethnicity records.	Fair
Marazzato et al. [12]	Case control	Evaluate the microbial composition in the anterior nares of pediatric subjects suffering from AR, AH or both diseases	AR: 19 AH: 20 Both: 13 HC: 13	6–12 years	Unknown, Italy	Anterior nasal swab	16S rRNA amplification of the V3–V4 region	Children with AR and AH show similar alterations in nasal microbiota. Healthy nasal microbiota: abundance of *Moraxella nonliquefaciens* and *Corynebacterium pseudodiphtericum*. AR and AH nasal microbiota: abundance of *Acinetobacter guillouiae*, *A. gerneri* and *Pseudomonas brenneri*.	Small sample size. No ethnicity records.	Good
Morin et al. [17]	Prospective cohort	Evaluate the development of AR at age 6 years based on early-life microbial exposures	1 week: 29 AR vs. 332 HC 1 month: 38 AR vs. 403 HC 3 months: 38 AR vs. 407 HC	1 week, 1 month, and 3 months	European ancestry, Denmark	Fluid aspirated with a soft catheter passed through the nose to the hypopharyngeal region	16S rRNA amplification of the V4 region	Early-life nasal microbiome diversity is lower in children who develop AR by age 6 years. 1 week: abundance of *Streptococcus* and *Veillonella*. Relationship between DNAm and microbial diversity only at 1 week, but not at the other time points or with other diversity metrics.	V4 region is considered a relatively low informative region for taxonomic assignment. DNAm profiles and microbiome composition measured in different upper airway niches (inferior turbinate vs. hypopharynx, respectively)	Good
Ta et al. [18]	Case control	Evaluate the development of the nasal microbiota with AR and wheeze over 7 time points (3 weeks and 3, 6, 9, 12, 15, and 18 months) in the first 18 months	AR with wheeze: 34 AR without wheeze: 28 HC: 60	3 weeks and 3, 6, 9, 12, 15, 18 months	Chinese, Malay, and Indian, Singapore	Anterior nasal swab	16S rRNA amplification of the V3–V6 region	Nasal microbiome diversity HC: ↑ over time. Nasal microbiome diversity both AR groups: ↓over time. Although differed in bacterial composition. AR: ↑ in abundance of *Oxalobacteraceae* and *Aerococcaceae*. HC: ↑ in abundance of *Corynebacteriaceae* and early colonization with the *Staphylococcaceae*. Nasal microbiome is involved in development of early-onset rhinitis and wheeze in infants.	Difficulties in identification all bacterial groups down to the species level.	Good
Yau et al. [14]	Cross-sectional	Evaluate the changes in nasal and ocular surface microbiome with ARC	ARC: 23 HC: 17	6–18 years	Unknown, China	Nasopharyngeal nasal swab	16S rRNA amplification of the V3–V4 region	ARC: Nasal microbiome ∼ ocular microbiome, but ≠ in HC. Most abundant genus: *Moraxella* (HC 53%; ARC 32%), *Corynebacterium* (HC 21%; ARC 13%), *Dolosigranulum* (HC 11%; ARC 11%), *Haemophilus* (HC 8%; ARC 5%), *Streptococcus* (HC 3%; ARC 7%), *Staphylococcus* (HC 2%; ARC 14%)	Small sample size. No ethnicity records.	Good

## Data Availability

Not applicable.

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
