# Peer review of "Microbiome in Nasal Mucosa of Children and Adolescents with Allergic Rhinitis: A Systematic Review"

_children, 2023, doi:10.3390/children10020226_

Round 1

Reviewer 1 Report

This systematic review represents the first report regarding the alterations in the na sal mucosa microbiome in children and adolescents suffering from AR.

The analysis of  the selected studies provides evidence of limited knowledge on this research field in pediatric age.

This suggests that a dysbiosis in nasal mu-399 cosa may impact children health, by rising the prevalence of chronic respiratory conditions (e.g., asthma, pneumonia).

These studies are particularly important, to help out clinicians getting a better understanding of nasal microbiome and allergic inflammation, as  well as offer insights for future development of potential biomarkers for AR diagnosis and  its treatment. 

Author Response

Reply: Thank you for the comments, the effort and the time taken to review the manuscript. Thank you for pointing out the strengths of this study.

Reviewer 2 Report

This is a very significant systematic review and an interesting read.However, there are some concerns in the method. Improvement in these areas would make for a better systematic review.

1.There are concerns about the comprehensiveness of the literature search. What are the reasons for not searching the Cochrane Library?

2. Did you pre-register for a systematic review protocol? Insufficient description of inclusion and exclusion criteria, and exclusion for not being in the English literature, but questions remain about the legitimacy of the reasons for exclusion.

Overall, there is a concern that quality is not sufficiently ensured in the process of selecting literature and determining search keywords. We are also concerned about whether advice from experts (e.g., librarians) was obtained in setting keywords. These concerns could be remedied by detailed description in the methodology. In conducting a systematic review, comprehensiveness of literature selection is important, and the authors should give more consideration to these issues.

Author Response

This is a very significant systematic review and an interesting read. However, there are some concerns in the method. Improvement in these areas would make for a better systematic review.

Reply: Thank you for the comments, the effort and the time taken to review the manuscript. All Materials and Methods section was thoroughly and carefully revised as suggested.

1.There are concerns about the comprehensiveness of the literature search. What are the reasons for not searching the Cochrane Library?

Reply: We performed an electronic search on 4 databases, such as MEDLINE (PubMed), Web of Science, Scopus, and Google Scholar. Despite Cochrane Library (Cochrane reviews) are gold standard quality, is a mainly resource for systematic reviews; published articles taken from MEDLINE and EMBASE; methods used in the conduct of controlled trials; and abstracts of systematic reviews. The goal of the current study was search and compare the results from original studies (scientific papers) regarding the microbiome of nasal mucosa of children and adolescents with AR, excluding reviews.

2. Did you pre-register for a systematic review protocol? Insufficient description of inclusion and exclusion criteria, and exclusion for not being in the English literature, but questions remain about the legitimacy of the reasons for exclusion.

Reply: We register our systematic review protocol in PROSPERO database, with a provisional registration number: ID 390751). We added this information in Line 87-89.

The exclusion criteria were determined after setting the research goals for this review. Considering that our review aimed to search for original studies from scientific papers, we excluded other studies whose outcomes could not be of interest for this work, for being self-reported or not using objective measures. Regarding the exclusion of articles written in languages other than in English, we used language restriction only at the stage of selecting, but not at the stage of searching the literature. The exclusion of non-English studies was mainly to avoid introducing bias in this review and ignoring key data due to misunderstanding translations. The inclusion and exclusion criteria section were improved as suggested and marked in the manuscript.

Overall, there is a concern that quality is not sufficiently ensured in the process of selecting literature and determining search keywords. We are also concerned about whether advice from experts (e.g., librarians) was obtained in setting keywords. These concerns could be remedied by detailed description in the methodology. In conducting a systematic review, comprehensiveness of literature selection is important, and the authors should give more consideration to these issues.

Reply: We used keywords (alone or in combination) related to scientific literature concerning the microbiome of nasal mucosa of children and adolescents with AR. For this research, we used the MeSH Browser from the NIH (National Institute of Health) that allows to search directly for MeSH terms. We added this information and the query used in detail in the methodology as suggested.

Reviewer 3 Report

Research trends in the last decade have brought the human microbiota in the spotlight, owing to its role in homeostasis, defense mechanisms and possible prognostic role for a plethora of diseases. in this paper, the Authors present a review investigating possible upper airways microbiota phenotypes relating to allergic diseases. While there is a certain scarcity of literature data pertaining to this specific topic, significant and detectable changes in upper airways microbiota might constitute a simple, effective and well-accepted way to predict the development of AR in children.

I find the Authors' work to be scientifically adequate, well written and of interest. Only a few rather minor remarks:

- Figure 2: I would suggest declaring the HC and AR abbreviations in the caption aswell.
- Line 203: "descrease" - typo 

Author Response

Research trends in the last decade have brought the human microbiota in the spotlight, owing to its role in homeostasis, defense mechanisms and possible prognostic role for a plethora of diseases. in this paper, the Authors present a review investigating possible upper airways microbiota phenotypes relating to allergic diseases. While there is a certain scarcity of literature data pertaining to this specific topic, significant and detectable changes in upper airways microbiota might constitute a simple, effective and well-accepted way to predict the development of AR in children.

Reply: Thank you for the comments, the effort and the time taken to review the manuscript. Thank you for pointing out the strengths of this study.

I find the Authors' work to be scientifically adequate, well written and of interest. Only a few rather minor remarks:

- Figure 2: I would suggest declaring the HC and AR abbreviations in the caption as well.

Reply: Revised as suggested.

- Line 203: "descrease" – typo

Reply: Revised as suggested.

Reviewer 4 Report

I appreciate the opportunity to review the manuscript for publication in MDPI Children.

I feel that the topics are interesting and the manuscript is grossly well organized.

I have one comment.

File S1. Search strategy.

This figure is difficult to understand. It should be arranged as a table and cited reference numbers.

Author Response

Reply: Thank you for the comments, the effort and the time taken to review the manuscript. Thank you for pointing out the strengths of this study. We removed the File S1 regarding the search strategy and added this information that contain the query used for searching the literature in materials and methods section.

Round 2

Reviewer 2 Report

I have reviewed the revised manuscript. The authors were able to answer our questions appropriately. Thank you very much.